# Influence of Molecular Structure and Physicochemical Properties of Immunosuppressive Drugs on Micelle Formulation Characteristics and Cutaneous Delivery

**DOI:** 10.3390/pharmaceutics15041278

**Published:** 2023-04-19

**Authors:** Julie Quartier, Maria Lapteva, Younes Boulaguiem, Stéphane Guerrier, Yogeshvar N. Kalia

**Affiliations:** 1School of Pharmaceutical Sciences, University of Geneva, CMU-1 rue Michel Servet, 1211 Geneva, Switzerland; 2Institute of Pharmaceutical Sciences of Western Switzerland, University of Geneva, CMU-1 rue Michel Servet, 1211 Geneva, Switzerland; 3Geneva School of Economics and Management, University of Geneva, 40 Boulevard du Pont d’Arve, 1204 Geneva, Switzerland

**Keywords:** immunosuppressive drugs, molecular similarity, physicochemical parameters, polymeric micelles, skin topical delivery, TPGS

## Abstract

The aim of this study was to investigate whether subtle differences in molecular properties affected polymeric micelle characteristics and their ability to deliver poorly water-soluble drugs into the skin. D-α-tocopherol-polyethylene glycol 1000 was used to prepare micelles containing ascomycin-derived immunosuppressants—sirolimus (SIR), pimecrolimus (PIM) and tacrolimus (TAC)—which have similar structures and physicochemical properties and have dermatological applications. Micelle formulations were prepared by thin-film hydration and extensively characterized. Cutaneous delivery and biodistribution were determined and compared. Sub-10 nm micelles were obtained for the three immunosuppressants with incorporation efficiencies >85%. However, differences were observed for drug loading, stability (at the highest concentration), and their in vitro release kinetics. These were attributed to differences in drug aqueous solubility and lipophilicity. Differences between the cutaneous biodistribution profiles and drug deposition in the different skin compartments pointed to the impact of differences in thermodynamic activity. Therefore, despite their structural similarities, SIR, TAC and PIM did not demonstrate the same behaviour either in the micelles or when applied to the skin. These outcomes indicate that polymeric micelles should be optimized even for closely related drug molecules and support the hypothesis that drugs are released from micelles prior to skin penetration.

## 1. Introduction

The ability of polymeric micelles to encapsulate and to deliver poorly water-soluble drugs efficiently to the skin has made them of increasing interest for dermatological applications [1,2,3]. These aqueous colloidal systems offer a promising alternative to classical approaches used to formulate drugs with physicochemical properties that are sub-optimal for cutaneous delivery. The successful formulation of a drug in polymeric micelles is closely related to the drug-polymer affinity. Whereas the determinant factors influencing the incorporation of drugs in micelles have been investigated, the parameters affecting drug release and delivery to the skin have been less explored.

Considering that molecules prescribed to treat skin diseases may have similar structures and physicochemical properties, a rational approach in the selection of the polymer excipient to formulate polymeric micelles would help to overcome the high cost and time-consuming effort generated during the drug development process. In fact, in drug discovery, molecular similarity is one of the key concepts in the identification of new compounds. This drug design is based on the principle that molecules with molecular similarities are more likely to exhibit similar properties [4]. Therefore, the investigations of the influence of the physicochemical properties of structurally similar drugs on polymeric micelle formulation and, subsequently, on drug delivery in the skin would facilitate the transition from the drug discovery to the drug development process. While computational approaches already studied this relationship, in vitro experimental outcomes are still missing [5].

One example of molecular similarity in the dermatology area is the family of immunosuppressive drugs, which are widely prescribed to treat autoimmune diseases (Figure 1). Ascomycin, the parent compound of these molecules, was discovered in a soil sample collected from Easter Island and isolated from Streptomyces hygroscopicus. Ascomycin was at first recognized for its antifungal properties and its immunosuppressive activities were identified with the discovery of sirolimus (SIR) and tacrolimus (TAC) [6]—natural products isolated from Streptomyces hygroscopicus and Streptomyces tsukubaensis, respectively [7,8]. On the other hand, pimecrolimus (PIM) is a chemical analog that was synthesized from ascomycin. In fact, once the bioactivities of ascomycin were fully understood, extensive efforts were made to identify an alternative therapy to corticosteroids to treat inflammatory skin diseases and PIM was selected for its favorable safety and pharmacology profiles [9,10]. Despite their different origin, the three immunosuppressive drugs are structurally similar macrolides (Figure 1; Table 1).

Different approaches exist to estimate the similarity between molecules. One of them is to compare the molecules in terms of constitution, meaning the similarity in the backbone and the functional groups [11]. As highlighted in Figure 1, the difference between PIM and TAC molecules lies in two different functional groups: in position C-21, an ethyl chain is present for PIM and an allyl chain for TAC. Moreover, in position C-32, the functional group is a chlorine and a hydroxyl group for PIM and TAC, respectively. SIR exhibits major differences, especially with the triene double bond in position from C-17 to C-22, increasing the SIR molecular weight.

In addition to these molecular similarities, Table 1 summarizes the physicochemical properties, the mechanism of action and the indications of the three immunosuppressive drugs. SIR, PIM and TAC are characterized by a high molecular weight, a moderate lipophilicity and a poor aqueous solubility. Moreover, the three drugs have similar bioactivities—their mechanism of action involves a decrease of T-cell proliferation through the binding to a cytosolic protein, FK-binding protein 12. However, depending on the molecule, the complex interacts with different targets and at different phases of T-cell activation [10,12]. Hence, PIM and TAC inhibit calcineurin, while SIR inhibits the mTOR. Regarding their dermatological indications—TAC (Protopic^®^; 0.03 and 0.1% *v*/*v*) and PIM (Elidel^®^; 1% *v*/*v*) are prescribed as topical treatment for atopic dermatitis. Moreover, TAC is also used off-label to treat moderate psoriatic lesions. Contrary to the two other molecules, SIR has no approved indication for skin diseases but numerous case studies have recently investigated the role of SIR in the treatment of facial angiofibromas.

**Table 1 pharmaceutics-15-01278-t001:** Properties of immunosuppressive drugs.

	Sirolimus	Pimecrolimus	Tacrolimus
**Physicochemical properties**			
Molecular weight (g/mol) *	914.2	810.4	804.0
logP *	4.17	4.31	3.23
Aqueous solubility **	Poor	Poor	Poor
Ionizable	No	No	No
**Mechanism of action**			
	Inhibitor of mTOR target [13]	Inhibitor of calcineurin target [14]	Inhibitor of calcineurin target [14]
**Indications**			
Oral Topical	Graft rejection Facial angiofibromas (off-label) [15,16], Psoriasis (off-label) [17]	- Atopic dermatitis, Psoriasis (off-label) [18]	Graft rejection Atopic dermatitis, Psoriasis (off-label) [19]

* Predicted with Chem 3D 16.0, ** Measured experimentally (values presented below in Section 3).

We have previously used polymeric micelles to create aqueous formulations of poorly water-soluble drugs for cutaneous delivery in order to improve local bioavailability [1,2,20,21,22,23,24,25,26]. Given the structural similarities and the similar physicochemical properties of SIR, PIM and TAC, including their poor aqueous solubility, they were selected to investigate whether the small differences between them would be sufficient to alter the characteristics of the drug-loaded polymeric micelles made using D-α-tocopherol-polyethylene glycol 1000 (TPGS), and how they would affect cutaneous drug delivery and biodistribution. Molecule-dependent differences in drug delivery would lend further support to the hypothesis that drugs were released from the micelles prior to skin entry and that intact micelles did not penetrate bulk skin. Therefore, the specific objectives were: (i) to prepare micelles loaded with SIR, PIM and TAC at different concentrations, (ii) to exhaustively characterize the micelle formulations in terms of drug content, size, morphology, stability and in vitro drug release, (iii) to study the cutaneous delivery and biodistribution from the optimized micelle formulations, and (iv) to identify differences between the results obtained with the different drug molecules and to explain them.

## 2. Materials and Methods

### 2.1. Materials

PIM, TAC, and SIR were purchased from Hangzhou Dayang Chem (Hangzhou, China) and SIR-D3 (917.2 g/mol) from TLC Pharmaceutical Standards (Aurora, ON, Canada). Acetronitrile (ACN) (HPLC grade and LC/MS grade) was received from Fisher Scientific (Reinach, Switzerland) and triethylamine from Fluka (Buchs, Switzerland). TPGS (1513 g/mol) (Figure 2), isopentane, Dulbecco’s phosphate buffered saline (without calcium chloride and magnesium chloride; DPBS), and acetic acid were purchased from Sigma Aldrich (Buchs, Switzerland). Formic acid (extra-pure 99%) (FA) was obtained from Biosolve Chemicals (Dieuze, France). Brij™ C20-PA-(RB) was purchased from Croda Europe (East Yorkshire, England). Bovine serum albumin (BSA) was obtained from Axon Lab (Baden-Dättwil, Switzerland). O.C.T. mounting medium was provided by VWR Chemicals (Leuven, Belgium). Trifluoroacetic acid (extra-pure 99%) (TFA), acetone and Nile-Red dye were obtained by Acros Organics (Geel, Belgium). Ultrapure water (Millipore Milli-Q Gard 1 Purification Pack resistivity >18 MΩcm; Zug, Switzerland) with a filter (Millipak^®^ 40 Millipore) of 0.22 μm was used to prepare all solutions.

### 2.2. Analytical Methods

#### 2.2.1. Quantification by UHPLC-UV

The different UHPLC-UV settings for PIM, TAC and SIR are presented in Table 2. The UHPLC-UV analysis was performed by using a Waters Acquity UPLC^®^ system (Baden-Dättwil, Switzerland) with a quaternary solvent manager and sample manager and a TUV-detector (Baden-Dättwil, Switzerland). MassLynx software was used for integration and data analysis. The UHPLC-UV method was validated based on ICH and FDA Bioanalytical Method Validation guidelines [27,28].

#### 2.2.2. Quantification with UHPLC-MS/MS

UHPLC with tandem mass spectrometry detection (UHPLC-MS/MS) was used to quantify drug release and drug deposition in and permeation across skin during cutaneous delivery studies. The Waters Acquity UPLC^®^ system (Baden-Dättwil, Switzerland) comprised a binary solvent pump, a sample manager and a sample organizer, coupled to a Waters XEVO^®^ TQ-MS detector (Baden-Dättwil, Switzerland). Isocratic separation was carried out using a Waters Xbridge BEH C8 2.1 × 50 mm column containing 2.5 μm particles. For the analysis of PIM and TAC, the mobile phase consisted of a 10 mM ammonium acetate buffer with 0.01% acetic acid and acetonitrile (10:90 *v*/*v*). The flow rate was set at 0.55 mL/min and the column was thermostatted at 55 °C. For the analysis of SIR, the mobile phase consisted of ultrapure water with 0.1% triethylamine and acetonitrile (15:85 *v*/*v*). The flow rate was set at 0.5 mL/min with the column thermostatted at 45 °C.

Mass spectroscopy detection was performed with electrospray ionization using the multiple reaction monitoring (MRM) mode. MassLynx software was used for data integration and analysis. The UHPLC-MS/MS method was validated based on ICH and FDA Bioanalytical Method Validation guidelines (results presented in Appendix A) [20,21]. To compensate for the matrix effect, each injected sample contained an internal standard at a concentration of 90 ng/mL. PIM and TAC were respectively the internal standard for each other. For SIR, SIR-D3 was chosen as the internal standard.

The different UHPLC-MS/MS settings for PIM, TAC and SIR are presented in Table 3.

### 2.3. Preparation of the Micelle Formulation

#### 2.3.1. Thin-Film Hydration Method

Micelle formulations were prepared using the thin-film hydration method. Briefly, a known quantity of drug and TPGS was dissolved in 2 mL of acetone. The acetone was slowly removed by rotary evaporation (Büchi Vac V-513 Rotavapor^®^; Flawil, Switzerland) until the formation of a uniform thin-film, which was left under the hood overnight and then hydrated with 4 mL of ultrapure water. After equilibration for 2 h, the micelle solution was centrifuged at 10,000 rpm for 15 min (Eppendorf Centrifuge 5804; Hamburg, Germany) to remove excess drug, and the supernatant was carefully collected.

#### 2.3.2. Optimization of Drug and Polymer Content

The micelle formulation was optimized in order to reach a high drug loading. During screening studies, TPGS content was kept fixed at 50 mg/mL and different concentrations of drug (1, 1.25, 1.5, 2, 2.5 and 5 mg/mL) were added during the micelle formulation process. The formulation with the highest drug content, which satisfied reproducibility and stability criteria, was chosen for the subsequent in vitro release and skin delivery studies. Drug loading was determined by UHPLC-UV.

### 2.4. Characterization of Micelle Formulations

#### 2.4.1. Drug Solubility in Water and Aqueous Solutions of TPGS

Saturated solutions of SIR, PIM and TAC were prepared in water and in an aqueous solution of TPGS (50 mg/mL). The solutions were kept at room temperature during a 24 h period of stirring. The samples were then centrifuged and the supernatant was collected and diluted in ACN prior to a UHPLC-MS/MS analysis. All the samples were prepared in triplicate.

#### 2.4.2. Thermal Properties

Differential scanning calorimetry (DSC) was performed for the thermal investigation of the drugs and TPGS (DSC 3 STARe System, Mettler Toledo; Greifensee, Switzerland). Samples of pure drug, pure TPGS and physical mixtures were sealed in aluminum hermetic pans with a pierced lid. Each sample was heated from 0 to 200 °C at a heating rate of 5 °C/min under a nitrogen atmosphere at a flow rate of 80 mL/min. An empty sealed aluminum pan was used as reference.

#### 2.4.3. Size and Morphology Characterization

Micelle size was characterized by the hydrodynamic (Zav) and the number-weighted diameters (dn), measured using dynamic light scattering (DLS) with a Zetasizer Nano-ZS (Malvern Instruments Ltd.; Malvern, United-Kingdom) at an angle of 90° and at a temperature of 25 °C. All measurements were performed in triplicate.

Micelle morphology was analyzed by transmission electron microscopy (TEM) (FEI Tecnai™ G2 Sphera; Eindhoven, The Netherlands) using negative staining. Briefly, 5 µL of micelle solution was dropped onto an ionized carbon-coated copper grid (0.3 Torr, 400 V for 20 s). Then, the grid was put in contact with a 100 μL drop of a saturated uranyl acetate aqueous solution for 1 s and then for 60 s in another 100 μL drop. The excess staining solution was removed, and the grid was dried at room temperature prior to the measurement.

#### 2.4.4. Drug Content Determination

The amount of drug loaded in the micelles was quantified by UHPLC-UV. To ensure the complete destruction of the micelles and the release of encapsulated drug, samples were diluted in ACN prior to analysis.
(1)$$Drug\;content\left(\frac{{mg}_{drug}}{{mL}_{formulation}}\right)=\frac{mass\;of\;drug\;in\;the\;formulation\left(mg\right)}{volume\;of\;the\;formulation\left(mL\right)}$$
(2)$$Drug\;loading\left(\frac{{mg}_{drug}}{g_{polymer}}\right)=\frac{drug\;in\;the\;formulation\left(mg\;{mL}^{-1}\right)}{copolymer\;in\;the\;formulation\left(g\;{mL}^{-1}\right)}$$
(3)$$Incorporation\;efficiency\left(\%\right)=\frac{mass\;of\;drug\;incorporated\;in\;micelles\left(mg\right)}{mass\;of\;drug\;introduced\left(mg\right)}\times 100$$

#### 2.4.5. Stability

Stability of the micelle solution was evaluated after storage at 4 °C for 5 months. Drug content and micelle size were determined at a series of time points (day 1, and then after 1, 2, 3, 4, and 5 months).

#### 2.4.6. In Vitro Drug Release from the Micelles

For the in vitro drug release study, the selected dissolution medium consisted of an aqueous solution containing 1% Brij™ C20-PA-(RB). The optimal micelle solution—containing 0.2% of SIR, PIM and TAC—was prepared. 1 mL of micelle solution was dispersed in a dialysis bag (Spectra/Por^®^ 3, dialysis membrane MWCO 3500, width 18 mm) and put in 13 mL of dissolution medium. The samples were kept under stirring in a bath maintained at 32 °C during 24 h. Aliquots (1 mL) were withdrawn at predetermined times (0, 1, 2, 4, 6, 9, 12, and 24 h). Dissolution medium was added to maintain sink conditions. The aliquots were diluted with ACN prior to UHPLC-MS/MS. All samples were analyzed in five replicates.

### 2.5. Evaluation of Skin Delivery In Vitro

#### 2.5.1. Porcine Skin Preparation

Porcine ear skin was used for skin delivery studies and was supplied by a local abattoir (CARRE; Rolle, CH). Briefly, skin samples were processed with a Zimmer air dermatome (Münsingen, Switzerland) to obtain pieces with a thickness of ∼800 µm. Hair was removed carefully from the skin surface using clippers. Circular disks of 30 mm were then punched out (Berg & Schmid HK 500; Urdorf, Switzerland) and stored at −20 °C until use, for a maximum period of 3 months.

#### 2.5.2. Drug Delivery under Infinite Dose Conditions

Porcine skin samples were mounted in Franz diffusion cells (Milian SA; Meyrin, Switzerland) with a formulation contact area of 2 cm^2^. In the donor compartment, the optimal micelle solution 0.2% was applied under infinite dose conditions (500 µL/cm^2^) during 12 h. The receptor compartment was filled with 10 mL of phosphate buffered saline at pH 7.4 containing 1% BSA. The receiver phase was stirred at 250 rpm and maintained at 32 °C throughout the experiment. These conditions complied with the OECD guidelines [29]. After the experiment, 1 mL of this phase was withdrawn to quantify drug permeation. Samples were diluted in ACN to precipitate BSA. After centrifugation at 5000 rpm at 4 °C during 15 min, the permeation samples were analysed by UHPLC-MS/MS. Each skin sample was carefully cleaned with a cotton swab and a wash solution to remove the residual formulation from skin surface.

#### 2.5.3. Investigation of Drug Biodistribution Profile

At the end of the experiment, a small area, 0.8 cm^2,^ was punched out from the 2 cm^2^ skin samples. These skin discs were snap-frozen in isopentane cooled by liquid nitrogen, followed by cryotoming (Thermo Scientific CryoStarTM NX70; Reinach, Switzerland) to obtain two lamellae with a thickness of 20 µm and 19 lamellae with a thickness of 40 μm. These 21 skin slices enabled the amount of drug to be quantified as a function of position down to a depth of ~800 µm, encompassing the stratum corneum, viable epidermis and upper and lower dermis, respectively. Drug deposited in each lamella was extracted in 300 µL of ACN overnight with continuous stirring at room temperature. The extraction samples were centrifuged at 5000 rpm for 10 min and diluted prior to UHPLC-MS/MS analysis.

#### 2.5.4. Data Analysis

The statistical methodology is similar to the one presented in Quartier et al. (2019) [30]. It consists in a comprehensive analysis of the difference between the formulations that is two-fold: first, a quantification of the difference that considers the dependence structure of the lamellae throughout skin depth, and second, one that quantifies their marginal differences. The former is conducted through a multivariate approach based on the Mahalanobis distance between the mean vectors of the two formulations, which assesses whether they are significantly different for at least one skin slice. The marginal differences are established using a univariate inference approach based on the Mann–Whitney–Wilcoxon (MWW) test which assesses the differences of each component of the multivariate mean vectors. Similar to the ANOVA analysis, the Mahalanobis distance-based test has the advantage of considering the differences between the two formulations for all skin layers at once. On the other hand, the MWW approach is informative in that it provides information on the locations, the directions as well as the magnitudes of the potential differences.

More formally, let $\mu_{k}=\left[\mu_{1k},\mu_{2k},\ldots ,\mu_{Jk}\right]$ denote the mean vector of formulation k∈{1,2,3} where each component of the vector corresponds to the drug deposition for the skin lamella at depth j∈{1,…,J}. For simplicity, consider SIR, PIM and TAC to correspond to k equal to 1, 2 and 3 respectively.

The *p*-values are obtained by conducting hypothesis-testing procedures for each of the Mahalanobis distance-based tests and the MWW test. Taking SIR and PIM as illustrative examples, the hypotheses for the first test are expressed as follows:$$H_{0}\text{:}\;\mu_{1}=\mu_{2},\;\;H_{1}\text{:}\;\mu_{1}\neq \mu_{2},$$
and the hypotheses for the second test are given by:(4)$$H_{0}\text{:}\;\mu_{j1}=\mu_{j2},\;\;H_{1}\text{:}\;\mu_{j1}\neq \mu_{j2},\;\;for\;j\in \{1,\ldots ,J\}.$$

In the case of the MWW test, the *p*-values need to be corrected for the multiple comparison problem as the testing procedure is applied for each layer independently, which results in an undesired increase of type-I error with the number of conducted tests, J. We have therefore opted for the False Discovery Rate (FDR) correction [31] which assumes a conditional (positive) association for the *p*-values—a plausible assumption for the data at hand.

## 3. Results

### 3.1. Development of Micelle Formulations

#### 3.1.1. Drug Solubility in Water and in Aqueous Solutions of TPGS

The solubility of the drugs was first investigated in water and an aqueous solution of polymer (Table 4).

In water, PIM and TAC exhibited very similar and very low solubility, and SIR was the most soluble molecule. The addition of TPGS in the aqueous solution significantly increased the aqueous solubility of the three drugs and the differences between them. At 50 mg/mL (33 mM) of TPGS, the concentration used for micelle formulation, the solubility was highest for SIR, followed by TAC and then PIM (1214.10 ± 123.50, 636.74 ± 73.45 and 330.32 ± 31.22 µM, respectively).

#### 3.1.2. Optimization and Characterization of Micelle Formulation

During micelle preparation, TPGS concentration was kept fixed at 50 mg/mL and different concentrations of each drug were tested (Table 5). The amount of drug encapsulated was expressed and evaluated in mg/mL (1, 1.25, 1.5, 2, 2.5, 5 mg/mL). Moreover, to compare the encapsulation of the immunosuppressive drugs into the micelles, the different formulations were prepared and were characterized in terms of drug content, incorporation efficiency and size (Figure 3).

The six micelle formulations prepared with SIR, PIM and TAC displayed a high incorporation efficiency, >85% (represented by the columns; Figure 3). Consequently, the highest drug content (approximately 4.5 mg/mL) was obtained with Formulation 6 for the three immunosuppressive drugs (represented by the lines, Figure 3). The sizes (dn) of the micelles were measured and found to be approximately 8 nm with a low polydispersity (<0.3) for all the formulations (results provided in Appendix A; Appendix A).

For the micelle solutions with high drug content (Formulations 5 and 6), the stability in terms of drug content (mg/mL) was also followed for 5 months (Figure 4). The stability of Formulation 5 was similar when SIR, PIM or TAC was encapsulated into the micelles. The drug content was approximately 2.3 mg/mL, which corresponds to 2.5, 2.8 and 2.9 mM of SIR, PIM and TAC, respectively. For Formulation 6 with the highest drug content, the micelle solutions loaded with PIM and TAC remained stable during the 5 months, whereas-SIR loaded micelles became unstable after 4 months. Given that Formulation 5 was stable for the three drugs, it was chosen for the further in vitro studies.

Finally, the micelle morphology of Formulation 5 was studied by TEM analysis. As presented in Figure 5, the images revealed that in addition to the smaller spherical shapes, “worm-like” structures could also be visualized for the three drug-loaded micelles. These have been seen previously with SIR but were not always present when other molecules, e.g., adapalene [22], or terbinafine and econazole [32] were incorporated into TPGS micelles.

#### 3.1.3. DSC Analysis

The thermal investigations were performed by DSC. The physical state of each drug mixed with TPGS was also explored by analyzing the physical mixtures of the drug and the polymer, according to their proportion in Formulation 5. Figure 6 represents the thermograms obtained for each sample.

The endothermic points of pure SIR, PIM and TAC were 181.82 °C, 163.11 °C and 127.20 °C, respectively, corresponding to their melting points. The TPGS thermogram showed two endothermic points (33.83 °C and 39.49 °C), characteristic of its crystalline nature [33,34]. The two melting points were already reported in previous studies and may be attributed to the different isomers form of TPGS [35].

For the physical mixtures, all thermograms showed the two melting endotherms of TPGS and the disappearance of the drug endothermic peak, indicating that the three drugs are more likely to be in the amorphous state. However, the absence of the endothermic peaks may also be due to a low concentration of the drugs within the physical mixture or the dissolution of the drug in the melted TPGS during the heating process [36,37]. Thus, X-ray diffraction analysis would need to be performed to confirm the amorphous or molecular dispersed state of the molecules.

#### 3.1.4. Drug Release Kinetics from the Micelles

The optimized Formulation 5 was used for these experiments and its drug content was adjusted to 2 mg/mL, which corresponds to 2.2, 2.5 and 2.5 mM of SIR, PIM and TAC, respectively. In vitro drug release kinetics were studied for the three drug-loaded micelles. In Figure 7, the concentrations of the drug in the release medium are presented as µmol per liter to standardize the results obtained for each molecule.

Although the drug loading (mg_DRUG_/g_TPGS_) in the micelles was similar for the three drugs, the release kinetics were different. The amount of SIR and TAC released from the micelles was significantly higher than PIM release. Moreover, PIM release had a linear profile, whereas for SIR and TAC, the release reached a plateau between 4 and 9 h, before increasing after 24 h.

### 3.2. Skin Delivery from Micelle Formulations 0.2% under Infinite Dose Conditions

Drug delivery studies were conducted using porcine skin. Infinite dose experiments (500 µL/cm^2^) were performed with micelle solution 0.2% applied during 12 h. The cutaneous biodistribution profiles were constructed and compared for the three drugs.

PIM exhibited the highest cutaneous bioavailability (PIM—287.7 ± 102.4 pmol/cm^2^) in comparison to SIR and TAC, for which the cutaneous delivery was more similar (SIR—155.5 ± 46.3 and TAC—134.8 ± 63.2 pmol/cm^2^, respectively).

Figure 8 represents the deposited amount of the drugs (in pmol/cm^2^) in each 40 µm-thick lamella as a function of depth. This data representation was first described in our previous publication into the cutaneous delivery of SIR from TPGS micelles [21] introducing the use of the Mahalanobis *p*-value as a method to determine difference between formulations when analyzing cutaneous biodistribution data. As explained in that manuscript, head-to-head comparisons using t-tests were inappropriate since the amounts present in each successive layer were dependent on the amount in the preceding layer. Thus, a new multivariate approach was required to address the codependency of the data. In brief, each panel in Figure 8A–C—presents a comparison of the results obtained for a given pair of molecules, with one considered to be the “reference” for the comparison. Thus, in Figure 8A, SIR (reference) is compared to TAC: (i) the plot on the left displays the “envelope” or range of the amounts of each molecule determined as a function of depth in the skin; (ii) the second plot displays the individual cutaneous biodistribution profiles for each replicate of that molecule (pale line in the respective color, e.g., in Figure 8A, fine green lines for SIR and fine blue lines for TAC) together with the mean value (thicker line in the respective color, e.g., in Figure 8A, thick green line for SIR and thick blue line for TAC)—note that the amounts are presented using a logarithmic scale which is necessary for the next plot; (iii) the third plot shows the mean log difference in the amounts present at each depth and presents the difference between the “treatment” and the reference—again, using Figure 8A as the example, this shows the difference between TAC and SIR. These differences are expressed as a mean and a confidence interval (with the mean difference expressed using a color-coded scale).

For SIR and TAC, the difference in drug deposition is graphically visible in the upper-dermis (200–360 µm), even though this difference is not statistically relevant when SIR and TAC depositions in each 40-µm thick skin lamellae are compared. In comparison to TAC and SIR, PIM deposition is significantly higher in the upper region of the skin (20–180 µm), corresponding to the epidermis.

## 4. Discussion

### 4.1. Development of Micelle Formulations

#### 4.1.1. Aqueous Solubility of the Drugs

TPGS is a non-ionic water-soluble derivative of vitamin E conjugated with polyethylene glycol. It is characterized by a MW of 1513 g/mol, an HLB of 13.2 and a CMC of 0.02% (0.2 mg/mL) [27]. TPGS was chosen to formulate the polymeric micelles because it has been approved by the U.S. Food and Drug Administration and European Medicines Agency as a pharmaceutical ingredient. Over these past years, the role of TPGS in drug delivery has been studied for various applications in nanomedicine (prodrugs, micelles, liposomes) to enhance drug solubility, permeability and stability [38,39]. Moreover, TPGS-based micelle formulations have already demonstrated their ability to enhance the solubility of sirolimus and other poorly water-soluble drugs (e.g., adapalene) and their efficacy to increase drug deposition in the skin compared to classical formulations [21,22]. For these reasons, TPGS was selected to investigate the physicochemical properties affecting the encapsulation and skin delivery of structurally similar molecules.

The solubility of the three drugs in water and in aqueous solutions of TPGS was investigated and demonstrated the first differences. Although the drugs are known to be poorly water-soluble, SIR displayed a significantly higher solubility (8.24 ± 0.22 µM) compared to PIM and TAC (0.17 ± 0.02 and 0.20 ± 0.05 µM, respectively). The addition of 50 mg/mL of TPGS significantly improved the drug water solubility, due to its surfactant role (147-fold for SIR, 1943-fold for PIM and 3184-fold for TAC). This preliminary study gave the first information about drug aqueous solubility and the interaction between the drugs and the polymer. It was expected that these results would be reflected during studies with the micelle formulations.

#### 4.1.2. Micelle Formulations Characterization

To fully investigate the encapsulation of the drugs into the polymeric micelles, six different formulations were prepared, for which TPGS polymer content was fixed at 50 mg/mL (5% *w/w*) and the drug content was varied from 1 to 5 mg/mL. Consequently, the drug loading was screened from 20 to 100 mg_DRUG_/g_TPGS_. The six micelle solutions could be formulated for the three immunosuppressive drugs with an incorporation efficiency higher than 85% (Figure 3). Moreover, the size of the micelles was not influenced by the nature of the drug loaded and was measured to be ~8 nm. It was also shown that the process to formulate the polymeric micelles increased significantly the water solubility of the three drugs—approximately up to 4.5 mg/mL (corresponding to 4.9 mM for SIR and 5.6 mM for TAC and PIM), in comparison to a simple addition of the drug in the aqueous solution of TPGS (1.2, 0.6 and 0.3 mM for SIR, TAC and PIM, respectively). Indeed, during the thin-film hydration method, the drug is first homogeneously dispersed in the film constituted of TPGS that will be subsequently hydrated with ultrapure water to promote the drug loading into the micelles [40].

However, when the stabilities of Formulation 5 and Formulation 6, which had the higher drug content (2.5 and 5 mg/mL, respectively), were monitored, it was seen that Formulation 5 remained stable for at least 5 months for the three drugs, whereas some differences were noticed between the drugs with respect to the stability of Formulation 6. In fact, Formulation 6 with SIR-loaded micelles became unstable after 4 months, which was not the case for PIM and TAC. The observations made with Formulation 6 did not correlate with the results obtained during the solubility study, with SIR having the highest solubility in the aqueous solution of TPGS. Moreover, when the drug content is expressed in moles per liter, Formulation 6 incorporated a targeted amount of drug corresponding to 5.5 mM for SIR and 6.2 mM for PIM and TAC. Therefore, although a lower molar concentration of SIR was encapsulated, this micelle formulation was the least stable.

The analysis by DSC could not demonstrate a difference in drug-polymer interactions (Figure 6). In fact, the physical mixtures of the three drugs with the polymer showed that the drugs were fully dissolved within TPGS. However, it was difficult to attest whether the drugs were in an amorphous state or simply dissolved in the melted TPGS matrix during the analysis. Therefore, to help in the understanding of the parameters influencing the drug encapsulation, the steric and physicochemical properties were detailed for each drug (Figure 9).

Due to the molecular similarities, the three immunosuppressive drugs exhibit a comparable distribution of lipophilic surface (Figure 9). However, since PIM contains a chlorine atom, the molecule has a higher lipophilic surface in the “head region”. It is also reflected in the logP value, with PIM having the highest logP: 4.31. This lipophilicity should favor drug encapsulation into the micelle core, constituted by the hydrophobic segment of TPGS (D-α-tocopherol). On the other hand, polar surface area (PSA) is lower for PIM (158 Å^2^), with PSA being generally inversely proportional to the logP of the molecules [41]. In contrast, SIR has the highest PSA (195 Å^2^), which could explain its higher aqueous solubility and higher affinity with the hydrophilic tail (PEG chain) but would entail a lower affinity with the hydrophobic core of TPGS and thus, a lower drug loading (Table 4). Moreover, the high MW of SIR (914.2 g/mol) could also limit the amount of drug loaded in the micelles.

It was previously demonstrated that the number of H bond donors and acceptors also plays a role in the encapsulation efficiency of the drugs. In fact, it was suggested that a greater number of H bond acceptors and donors may increase the drug-polymer affinity [1,42]. However, these assumptions could not be applied in the case of immunosuppressive drugs, as SIR has the highest number of H bond donors and acceptors, but lower micelle stability for a high drug loading.

Other factors such as the affinity of the drug with the polymer (measured with Flory-Huggins interaction parameter, χ), the interaction of the drug with the polymer (e.g., hydrophobic interaction, 𝜋-𝜋 stacking interaction, electrostatic interaction) and the volume of the hydrophobic core could also be studied to predict the drug-loaded amount [43,44,45]. However, the attribution of only one universal parameter to explain drug encapsulation efficiency is unrealistic [42,46]. This demonstrated that all the different parameters need to be taken into account to predict the drug incorporation into polymeric micelles.

#### 4.1.3. In Vitro Drug Release Profile

The concentration (µM) of SIR and TAC quantified in the release medium was significantly higher than for PIM (Figure 7). Considering the previous results obtained during the formulation characterization, i.e., higher aqueous solubility and lower micelle stability, it was expected that SIR would be released from the micelle at higher amounts and faster than the other immunosuppressive drugs. In fact, the drug release from the micelles also depends on the same parameters affecting drug encapsulation in the micelles [46,47]. Moreover, the slow drug release of PIM could be explained by its low aqueous solubility, delaying drug diffusion from the inner polymer matrix to the outer phase [48]. However, TAC also had a low aqueous solubility and its drug release was not delayed. Therefore, it seemed that, more than aqueous solubility, logP value of the drugs may also have an effect on the slow or fast release from the polymeric micelles. Indeed, hydrophobic drugs with a moderate logP such as PIM (4.31) seemed to have a slower release. On the other hand, TAC had the lowest lipophilicity (3.21) of the three drugs, which could explain its similar release profile to SIR.

Moreover, the drug release is also linked to the phase state of the molecule in the micelle—a drug that is not dissolved in the core compartment will display a delayed drug release [49,50]. Regarding the results from the DSC, it is more likely that the three molecules were dissolved in the TPGS matrix (Figure 6). In contrast, an excessive stabilization of the drug could also prevent fast drug release [49]. These factors are closely related to the localization of the drug in the micelle, either in the corona or in the core [49,51,52]. All these parameters demonstrated the complexity involved in predicting the behavior of the encapsulated drug.

### 4.2. In Vitro Skin Delivery

The determinant factors influencing the skin delivery of a drug encapsulated in micelles have remained relatively unexplored. Recently, polymeric micelles emerged in the search for innovative formulations to enhance the cutaneous drug bioavailability. In fact, it was demonstrated that molecules lacking the appropriate physicochemical properties for skin delivery may become good candidates once incorporated in polymeric micelles [1,2,3,22,26]. In addition to enhancing aqueous solubility, polymeric micelles also create a drug depot at the skin surface by increasing the number of molecules in contact with the skin and promote the accumulation of the drug in the hair follicles [2,22,24,25,26]. However, the mechanism of how micelles enhance delivery of drugs into the skin still needs to be investigated.

Considering our tenet that intact polymeric micelles, due to their size, do not penetrate the skin, cutaneous delivery is mainly influenced by the number of micelles in contact with the skin, the thermodynamic activity of the system and the physicochemical properties of the drug. Indeed, due to the heterogeneity of the skin, drug penetration is challenging. Firstly, the drug needs to partition from the micelle to the stratum corneum, considered as the skin barrier, and more specifically to the intercellular lipid matrix [53,54]. As was the case for the in vitro drug release study, the phase state of the molecule in the micelle is important in predicting drug partitioning. Then, the hydrophilic/lipophilic balance and the MW of the drug also affect drug diffusion in the different regions of the skin. Indeed, to diffuse passively through the stratum corneum, before partitioning in the viable epidermis, it is considered that molecule needs to have a particular range of parameters: low MW (<500 Da), intermediate lipophilicity (logP 1–3), limited hydrogen-bond-forming capacity and an adequate solubility [55].

Interestingly, the skin delivery study revealed different results from those predicted during the in vitro release study. Despite the slow release of PIM from the micelles, PIM exhibited the highest cutaneous bioavailability (PIM—287.7 ± 102.4, SIR—155.5 ± 46.3, TAC—134.8 ± 63.2 pmol/cm^2^, respectively). Moreover, the higher in vitro release of TAC and SIR from micelles was not reflected in skin delivery experiments. Indeed, the results of in vitro release studies can have little bearing on drug delivery and these skin delivery results highlighted the importance of studying the behavior of the micelle formulations directly in contact with the biological tissue.

To facilitate the interpretation of the results, Figure 10 represents, in percentage terms, the distribution of the total drug deposition in the different skin layers (stratum corneum, viable epidermis and dermis). It was first noticed that a high percentage of PIM was delivered in the stratum corneum. From this observation, it can be assumed that due to its higher lipophilicity, PIM has a partition coefficient more in favor of entry to the skin than TAC and SIR. Because the stratum corneum is known to be a “drug reservoir”, a larger amount of drug deposited in this anatomical region will be more likely to penetrate the viable epidermis. Actually, the percentage of PIM deposition was also higher in the viable epidermis in comparison to SIR and TAC. Interestingly, SIR and TAC had a large percentage of skin deposition in the dermis, even higher than in the viable epidermis for SIR. Hence, the affinity and the binding of the drugs with skin components could explain the difference in drug bioavailability.

It was already observed that PIM had a stronger affinity with the components of the upper layers of the skin compared to TAC [56]. In fact, although a similar binding was found for the protein corresponding to macrophilin-12 (also called FK-binding protein 12—the targeted cytosolic protein of immunosuppressive drugs), PIM showed a higher capacity-binding to other skin proteins. It is possible to assume that the stronger interaction with the skin components as well as the higher lipophilicity of PIM led to a higher drug deposition in the upper layers of the skin in comparison to TAC and SIR.

Moreover, the effect of the lipophilicity/hydrophilicity distribution on the skin delivery of PIM and TAC was also previously studied by Meingassner et al. [57] and Billich et al. [58]. It was shown that the permeation of PIM across the skin was lower than TAC, independent of the skin origin (porcine or human) and formulation composition. The difference in the permeation results was related to the lipophilicity of the drugs (logP of 4.31 and 3.23 for PIM and TAC, respectively). In fact, an increase of drug lipophilicity decreased the systemic exposure of the drug, as it was also demonstrated in vivo when the marketed products of PIM and TAC, Elidel^®^ and Protopic^®^ respectively, were compared [59]. In the present study, the effect of lipophilicity/hydrophilicity distribution was reflected with the lower percentage of PIM deposition in the dermis (known as a hydrophilic layer [60]) in comparison to TAC, which will consequently decrease the systemic exposure of the drug. In contrast, the considerable percentage of SIR deposited in the dermis can be related to its higher aqueous solubility. However, the permeation across the skin of SIR, TAC and PIM could not be compared since the concentration was under the limit of quantification for the three drugs (<3 ng/mL); contrary to the previous studies [57,58] the micelle solutions used here were applied for only 12 h instead of 48 h.

Finally, to go a step further in the investigation, the penetration pathway though the hair follicles could also be studied for the micelle formulations loaded with the three immunosuppressive drugs. Indeed, the outcomes of skin delivery might also be explained by a different preferential pathway of the drugs and their related micelle formulations.

### 4.3. Summary of the Comparative Studies

This work was conducted to help in the understanding of factors affecting the micelle formulation and the skin delivery of structurally similar drugs. Indeed, despite the preparation of standardized micelle formulations (the same amount of drug per g of polymer), the characteristics of formulations and the cutaneous bioavailability were not the same for the three immunosuppressive drugs—close structural similarities did not lead to similar micelle formulation properties and skin deliveries. Table 6 summarizes these different observations.

The molecular structure and the physicochemical properties of the immunosuppressive drugs were examined in detail to identify the determinant parameters. Although it was difficult to attribute one factor to the behavior of drug-loaded micelles, the aqueous solubility and the lipophilicity of the immunosuppressive drugs appeared to play a major effect. These findings demonstrated that, despite the high degree of similarity between the three drugs, each molecule is unique and it is unlikely that it will be possible to select a universal micelle formulation for all drugs.

## 5. Conclusions

This work had as an objective to investigate whether it was feasible to “standardize” polymeric micelle formulations of drugs with molecular similarities, with a view to facilitating the drug development process. In these studies, the focus was on the use of polymeric micelles to develop aqueous formulations of poorly water-soluble drugs for drug delivery into the skin. Throughout the investigations, the physicochemical parameters of the drugs were highlighted to explain the drug-polymer interactions that occurred during micelle formulation. Despite their structural and physicochemical similarities, SIR-, PIM- and TAC-loaded micelles were found to behave differently in the initial characterization studies.

Furthermore, it was also shown that the three formulations behaved differently in cutaneous delivery studies and that, once the drug was delivered into the skin, these factors also governed drug bioavailability and biodistribution. In the context of cutaneous drug delivery, the observation that the amounts of drug delivered and the biodistribution were different and drug-dependent, also supported the hypothesis that drugs are first released from the micelles, which disaggregate on contact with the skin, and do not themselves enter the stratum corneum (although they might accumulate in appendageal structures). At this point, the physicochemical properties of the “free” drug in the formulation determine its partitioning into the skin and subsequent transport.

These results suggested that it is not necessarily possible to assume that polymeric micelle formulations made using a given polymer of structurally related drugs with similar physicochemical properties will behave in an identical manner. Small variations in physicochemical properties will affect formulation characteristics and, in the case of topical drug delivery, bioavailability and biodistribution. However, it will be interesting to perform in vivo studies to confirm whether any differences found in vitro are again observed and whether they are clinically significant.

## Figures and Tables

**Figure 1 pharmaceutics-15-01278-f001:**
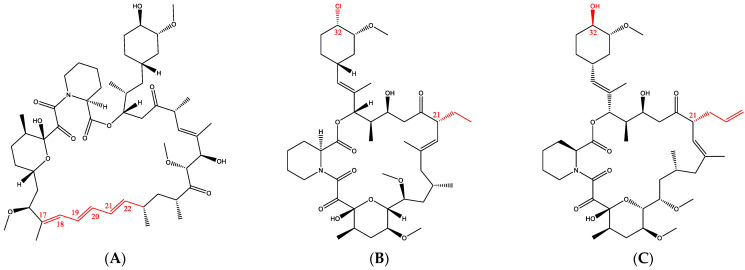
Structures of (**A**) sirolimus, (**B**) pimecrolimus and (**C**) tacrolimus.

**Figure 2 pharmaceutics-15-01278-f002:**
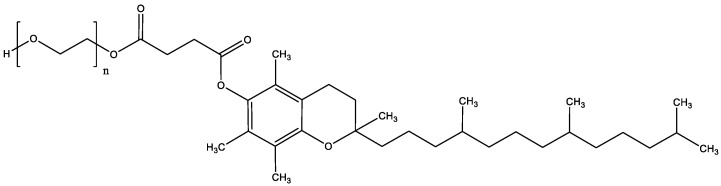
Structure of D-α-tocopherol-polyethylene glycol 1000.

**Figure 3 pharmaceutics-15-01278-f003:**
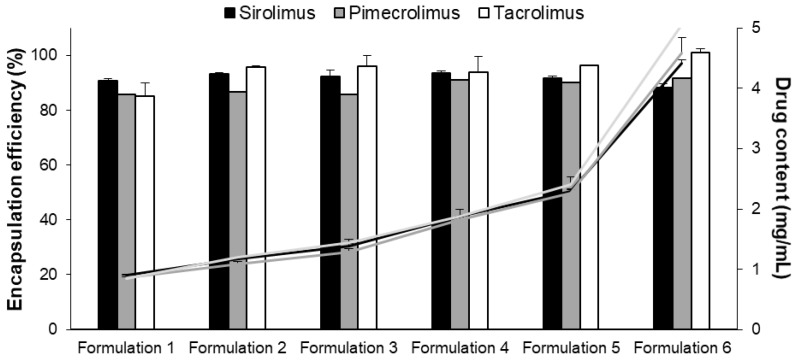
Encapsulation efficiency (%) (columns) and drug content (mg/mL) (lines) of the six micelle formulations. Tested groups: sirolimus (■ black), pimecrolimus (grey) and tacrolimus (white) micelle solutions (mean + SD, n = 3).

**Figure 4 pharmaceutics-15-01278-f004:**
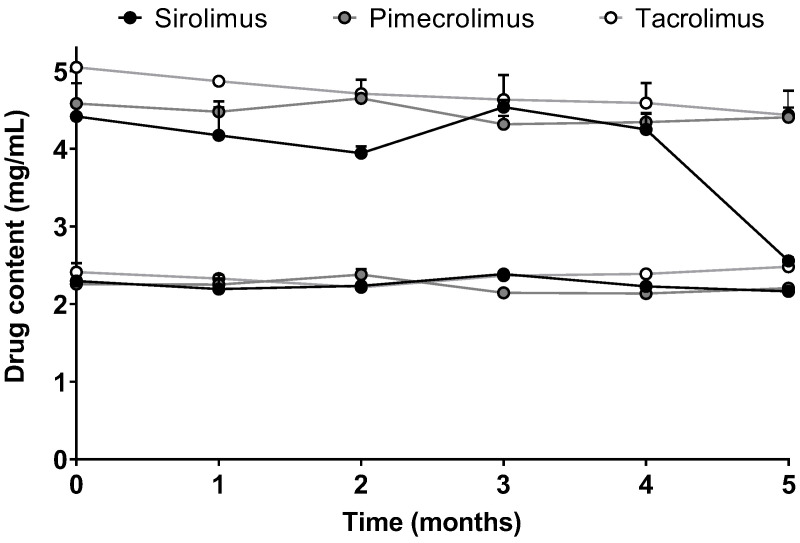
Stability of Formulation 5 and Formulation 6 in terms of drug content over time. Tested groups: sirolimus (black), pimecrolimus (grey) and tacrolimus (white) micelle solutions (mean + SD, n = 3).

**Figure 5 pharmaceutics-15-01278-f005:**
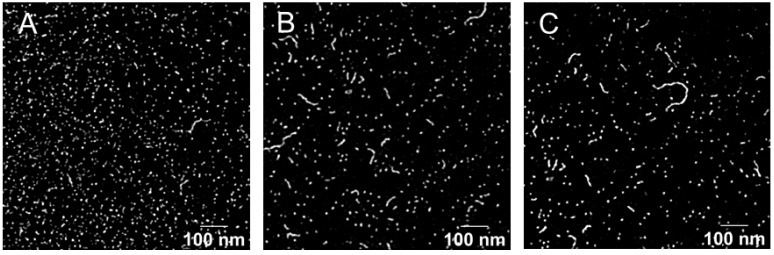
TEM image of (**A**) SIR, (**B**) PIM and (**C**) TAC loaded micelles. Scale bar (100 nm).

**Figure 6 pharmaceutics-15-01278-f006:**
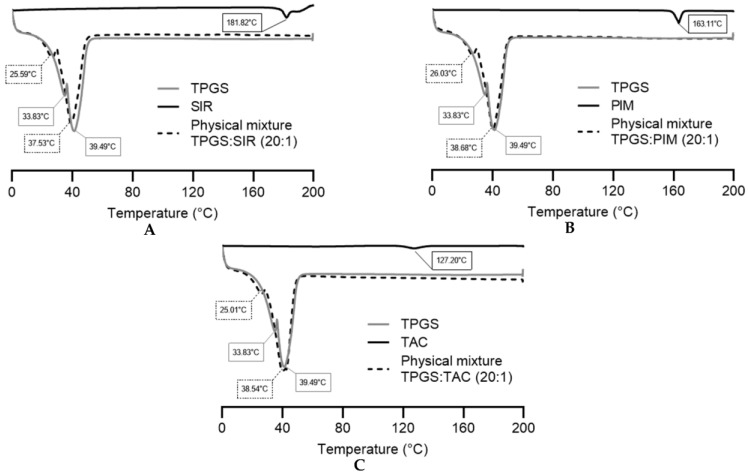
DSC thermograms of pure TPGS, pure drug and physical mixtures of (**A**) sirolimus, (**B**) pimecrolimus and (**C**) tacrolimus. Tested groups: TPGS (**—**), drugs (**—**) and physical mixtures (**- - -**).

**Figure 7 pharmaceutics-15-01278-f007:**
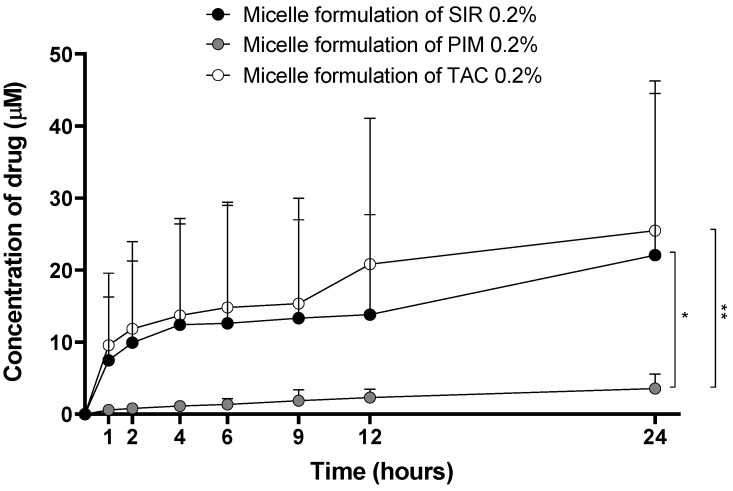
Drug release kinetics from the micelle formulations. Tested groups: sirolimus (black), pimecrolimus (grey) and tacrolimus (white) micelle solutions 0.2% (mean + SD, n = 5). *p*-values were calculated using Kruskal-Wallis rank test; statistical differences were denoted by asterisks (* *p* < 0.05, ** *p* < 0.005).

**Figure 8 pharmaceutics-15-01278-f008:**
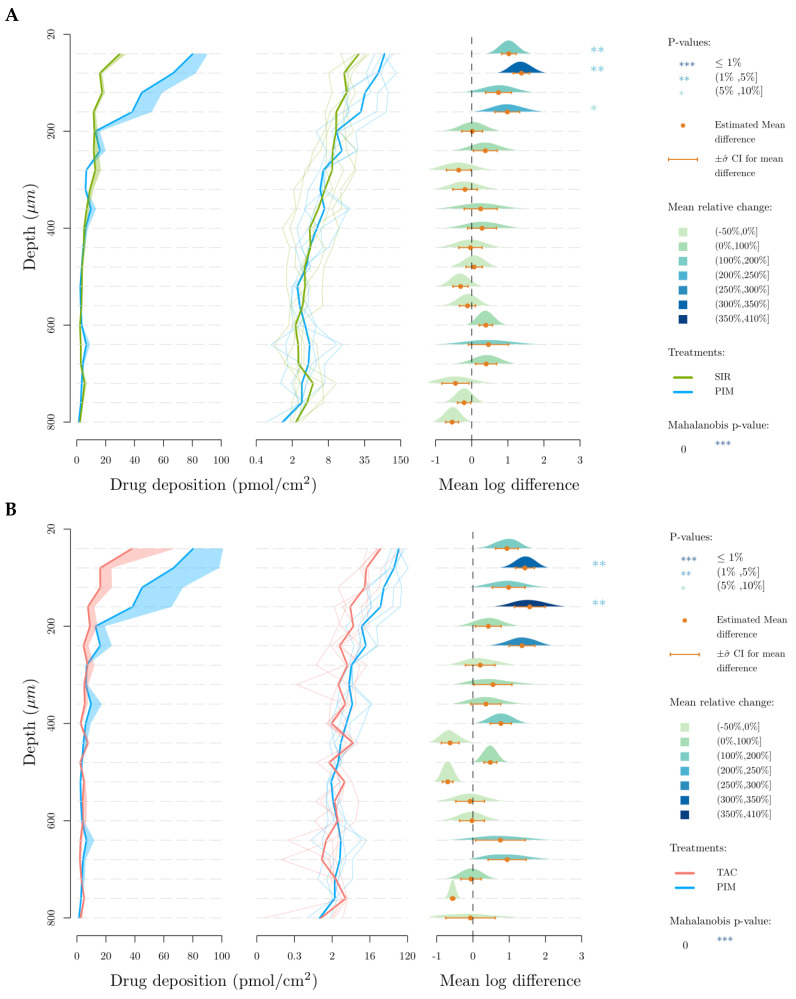
Cutaneous biodistribution profile of drugs in porcine skin lamellae (2 × 20 µm + 19 × 40 µm) to 800 µm (full depth) after 12 h of application time. Tested groups: (**A**) sirolimus (reference) vs. pimecrolimus, (**B**) tacrolimus (reference) vs. pimecrolimus and (**C**) tacrolimus (reference) vs. sirolimus. The first graph corresponds to the comparison of drug deposition in the original scale and the second in the log scale. The mean relative change corresponds to the increase of the second treatment relative to the reference treatment in the log scale. More specifically it corresponds, for a given layer, to the ratio between the mean treatment drug and the mean reference drug-1.

**Figure 9 pharmaceutics-15-01278-f009:**
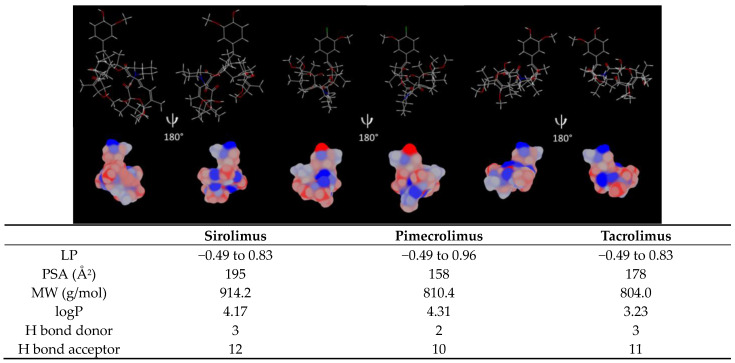
Sirolimus, pimecrolimus and tacrolimus lipophilic surface potential of the minimized energy conformation. The lipophilicity scale ranges from the most hydrophilic (blue) to most lipophilic (red) parts of the surface. LP: minimum/maximum lipophilicity values for each molecule entity, PSA: polar surface area, MW: molecular weight, logP and H bond: hydrogen bond acceptor and donor are listed in the table. Estimation given by MarvinSketch 20.9 and Chem 3D 16.0.

**Figure 10 pharmaceutics-15-01278-f010:**
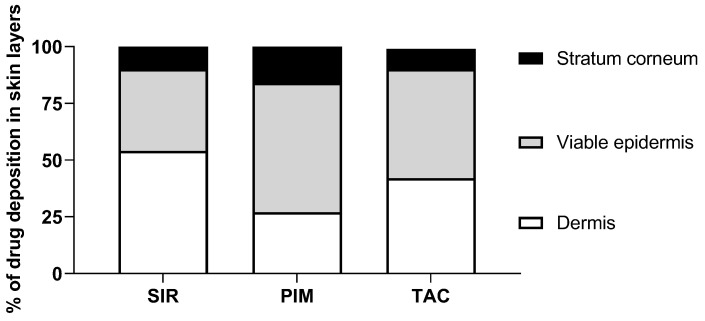
Drug deposited (%) in the three skin layers (stratum corneum, viable epidermis, dermis) from micelle formulation 0.2% of SIR, PIM and TAC.

**Table 2 pharmaceutics-15-01278-t002:** UHPLC-UV settings for detection of sirolimus, pimecrolimus and tacrolimus.

Parameters	SIR	PIM	TAC
Column	C8 2.5 μm, 2.1 × 50 mm	C18 2.5 μm, 2.1 × 100 mm	C18 2.5 μm, 2.1 × 100 mm
Mobile phase	(a) ACN and (b) Milli-Q water + 0.003% TFA (85:15 *v*/*v*)	(a) ACN and (b) Milli-Q water + 0.1% FA (95:5 *v*/*v*)	(a) ACN and (b) Milli-Q water + 0.1% FA (95:5 *v*/*v*)
Column temperature (°C)	45	45	45
Flow (mL/min)	0.5	0.5	0.5
Volume of injection (µL)	5	5	5
Wavelength (nm)	278	210	210
Retention time (min)	0.48	0.80	0.66
Limit of quantification (µg/mL) Limit of detection (µg/mL)	3.0 1.0	5.0 1.6	2.0 0.7

**Table 3 pharmaceutics-15-01278-t003:** MS7MS settings for detection of sirolimus, pimecrolimus and tacrolimus.

Parameters	SIR	SIR-D3	PIM	TAC
Nature of parent ion	[M − H]^−^	[M − H]^−^	[M + NH_4_]^+^	[M + NH_4_]^+^
Parent ion (*m*/*z*)	912.67	915.65	832.50	826.60
Daughter ion (*m*/*z*)	590.42	321.21	593.44	616.20
Collision energy (V)	36	40	30	40
Cone voltage (V)	60	52	60	50
Capillary voltage (kV)	3.6		3.0	
Source temperature (°C)	150		150	
Desolvation temperature (°C)	500		500	
Desolvation gas flow (L/h)	1000		850	
Cone gas flow (L/h)	0		0	
LM resolution 1	2.96		2.96	
HM resolution 1	15.00		15.00	
Ion energy 1 (V)	0.3		0.3	
LM resolution 2	2.91		2.91	
HM resolution 2	15.24		15.24	
Ion energy 2 (V)	0.6		0.6	
Limit of quantification (ng/mL)	3.0	/	3.0	3.0
Limit of detection (ng/mL)	1.0	/	1.0	1.0

**Table 4 pharmaceutics-15-01278-t004:** Solubility of drugs (µM) in water and in an aqueous solution of TPGS.

		Water	At TPGS Concentration (33 mM or 50 mg/mL)	Increase in Solubility
Drug solubility (µM)	SIR	8.24 ± 0.22	1214.10 ± 123.50	147×
	PIM	0.17 ± 0.02	330.32 ± 31.22	1943×
	TAC	0.20 ± 0.05	636.74 ± 73.45	3184×

**Table 5 pharmaceutics-15-01278-t005:** Composition of micelle formulations prepared with the thin-film hydration method.

Formulation									
	Target TPGS Content (mg/mL)	Target Drug Content (mg/mL)	Target Drug Loading (mg_DRUG_/g_TPGS_)	Drug Loading (mg_DRUG_/g_TPGS_)			Incorporation Efficiency (%)		
				SIR	PIM	TAC	SIR	PIM	TAC
Formulation 1	50	1.00	20	18.1 ± 0.2	17.2 ± 0.0	17.0 ± 1.0	90.7 ± 0.9	85.8 ± 0.0	85.1 ± 5.1
Formulation 2	50	1.25	25	23.3 ± 0.1	21.7 ± 0.0	23.9 ± 0.1	93.3 ± 0.4	86.7 ± 0.0	95.8 ± 0.5
Formulation 3	50	1.50	30	27.7 ± 0.8	25.8 ± 0.0	28.9 ± 1.1	92.2 ± 2.6	85.9 ± 0.0	96.1 ± 3.7
Formulation 4	50	2.00	40	37.5 ± 0.2	36.4 ± 0.0	37.6 ± 2.2	93.7 ± 0.6	91.0 ± 0.0	94.0 ± 5.5
Formulation 5	50	2.50	50	45.8 ± 0.4	45.1 ± 0.0	48.2 ± 0.0	91.7± 0.8	90.2 ± 0.0	96.4 ± 0.1
Formulation 6	50	5.00	100	88.4 ± 1.3	91.6 ± 0.1	101.0 ± 1.4	88.4 ± 1.3	91.6 ± 0.0	101.0 ± 1.4

**Table 6 pharmaceutics-15-01278-t006:** Summary of the comparative studies for the three drugs.

	Sirolimus	Pimecrolimus	Tacrolimus
MW	***	**	**
logP	**	***	*
Solubility in water	***	*	*
Solubility in aqueous solutions of TPGS	***	*	**
Stability in micelle formulation	**	***	***
In vitro drug release	***	*	***
Cutaneous drug delivery	*	***	*

* low, ** medium, *** high.

## Data Availability

The data presented in this study are available on request from the corresponding author.

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
