# Peer review of "Influence of Molecular Structure and Physicochemical Properties of Immunosuppressive Drugs on Micelle Formulation Characteristics and Cutaneous Delivery"

_pharmaceutics, 2023, doi:10.3390/pharmaceutics15041278_

Round 1
Reviewer 1 Report
In the paper the authors have investigated the influence of molecular characteristics of immunosuppressive drugs on TPGS micelles formulation and cutaneous delivery. The subject is worth of investigation. The paper is well organized and easy to follow, except for a figure that should be improved. The discussion is sensible and fully justified by the collected data. Methodologies are robust.
Minor comments:
2.3.2. Determination of drug and polymer content
Given the title, in this chapter the evaluation of drug and polymer content should be described. After centrifugation both drug and TPGS were quantified?
2.4.1. Drug solubility in water and aqueous solutions of TPGS
“Super-saturated solutions were prepared”. Please explain. How did you prepare these solutions? By simply adding an excess of drug to a TPGS solution? Why do you call them “supersaturated?”
Figure 5: TEM image show also worm-like structures. Do you think they are worm-like micelles or are simply an artifact linked to sample preparation?
Figure 7 : release. Another notable difference is the variability of the data, very low for PIM with respect to the other drugs. What could be the reason? What’s the % of micelles released after 24 h ?
Quality of Figure 8 should be improved. Additionally a more in depth explanation of the different panels should be given.In the first panel, what’s the meaning of the “shadow” close to the line? And why PIM/SIR/TAC profiles are different in panels A, B and C? What’s represented in the second panel?
I was wondering if the differences in skin accumulation (although not very high) could be ascribed also to a different “activity” of the drug in the formulation. i.e. to different loading capacities of the thin-layer-obtained micelles.
Reviewer 2 Report
The study was well-designed, and the results are detailed enough to examine the mechanisms behind the reported trends. There is a need to justify the following comments, in order to understand their practicability for the stated objectives.
Comments to the Author
The abstract is too long. Please concise the abstract and only provide the main findings.
Please provide any context in abstract for the significance of the study's findings, such as how they could inform the development of new formulations for dermatological or other applications.
The phrase "These outcomes suggested that the polymeric micelles need to be individually optimized for each immunosuppressive drug taking into account their individual physicochemical properties" could be rephrased for concision and clarity, such as "Optimizing polymeric micelles for each immunosuppressive drug may require accounting for their individual physicochemical properties."
The introduction should include research justification that led to the SIR, PIM and TAC.
The aims of the study are not clear. Please further elaborate on the aims and give more details.
The results and discussion do not go further than expected. It lacks a deepening. The results need proper discussion and comparison with previous similar studies to highlight the usefulness of the study.
Please include the appropriate statistical labeling in tables and figures where applicable.
Please make the digits in Figure 6 & 8 clearer as some of them are not visible
The novelty of the manuscript is questionable as already a lot of literature has been published. Please highlight the novelty of the manuscript
The conclusion needs to be improved. The conclusion is too general. What were the outcomes and prospects are not clear? The authors should justify elaborately how the current study is different from previous studies and useful to the scientific community.
